# EFFICIENT REPRESENTATIONS
# FOR PRIVACY-PRESERVING INFERENCE

## ABSTRACT

Deep neural networks have a wide range of applications across multiple domains such as computer vision and medicine. In many cases, the input of a model at inference time can consist of sensitive user data, which raises questions concerning the level of privacy and trust guaranteed by such services. Much existing work has leveraged homomorphic encryption (HE) schemes that enable computation on encrypted data to achieve private inference for multilayer perceptrons and CNNs. An early work along this direction was CryptoNets, which takes 250 seconds for one MNIST inference. The main limitation of such approaches is that of compute, which is due to the costly nature of the NTT (number theoretic transform) operations that constitute HE operations. Others have proposed the use of model pruning and efficient data representations to reduce the number of HE operations required. In this paper, we focus on improving upon existing work by proposing changes to the representations of intermediate tensors during CNN inference. We construct and evaluate private CNNs on the MNIST and CIFAR-10 datasets, and achieve over a two-fold reduction in the number of operations used for inferences of the CryptoNets architecture.

## 1 INTRODUCTION

In recent years, deep neural networks have achieved state-of-the-art accuracy for tasks such as image recognition. They have been deployed in a range of sectors, powering a wide variety of applications such as recommendation systems, medical diagnosis, and content filtering. Machine Learning as a Service (MLaaS) is a framework in which cloud services apply machine learning algorithms on user-supplied data to produce an inference result which is then returned to the user. Cloud systems are an attractive platform for deploying pretrained models due to the relatively low cost and the availability of remote servers. However, the data has to be decrypted before inference, which allows a server-side adversary to have access to the user's information. Homomorphic encryption (HE), can be applied to enable inference to be performed on encrypted data, enabling the result to be delivered to the user without risk of the server accessing the original data or the inference result. CRYPTONETS (Gilad-Bachrach et al., 2016) was the first application of HE to secure neural network inference, and leveraged the YASHE' scheme to perform MNIST classifications. CRYPTONETS suffers from a high number of homomorphic operations (HOPs), with a single MNIST inference requiring $\sim 290,000$ homomorphic multiplications and $\sim 250$ seconds of inference latency. Subsequent works such as FASTER CRYPTONETS (Chou et al., 2018) used neural network surgery and a faster encryption scheme to reduce the inference latency of CRYPTONETS.

Later works utilised ciphertext rotations as opposed to the SIMD packing scheme, enabling convolutional and fully connected layers to be computed using much fewer HOPs (Juvekar et al., 2018; Mishra et al., 2020). This has been shown to reduce the inference latency of MNIST models by more than an order of magnitude, bringing confidence that private inference can be practical. LOLA (Brutzkus et al., 2019) proposed novel representations for intermediate tensors and their MNIST model requires only 2.2 seconds for one inference. One drawback of their representations is poorly scalability to harder datasets such as CIFAR-10 due to the limited number of slots per ciphertext acting as a barrier to the size of tensors that are practical.

The limited set of operations supported by HE schemes prevents the secure computation of non-polynomial activation functions which impedes model training due to the problem of exploding

gradients (Chou et al., 2018). To address this, others have proposed the use of secure multi-party computation to enable secure computation of non-polynomial activations using multiple parties (Juvekar et al., 2018; Mishra et al., 2020). Despite enabling use of popular nonpolynomial activations such as ReLU, relying on multiparty computation incurs large amounts of data transfer between parties and the requirement for the parties involved to be online and have feasibly fast data transfer rates. For example, GAZELLE (Juvekar et al., 2018) requires ∼1 GB of data transfer per inference for their CIFAR-10 model, and DELPHI (Mishra et al., 2020) requires ∼2 GB per inference with a ResNet-32 model. Single-party approaches often choose to approximate the ReLU activation using a second-degree polynomial (Gilad-Bachrach et al., 2016; Chou et al., 2018; Brutzkus et al., 2019).

In this work, we introduce a framework for secure inference on secure CNNs, designed to reduce the number of HOPs required per inference whilst preserving prediction accuracy. Our contributions can be summarised as follows:

- We integrate the convolution-packing method from LoLa with the fast matrix-vector product method introduced by Halevi and Shoup (Halevi & Shoup, 2019) and utilised by Juvekar et al. (2018) in their multi-party computation framework. Intermediate convolutions are converted into fully-connected layers and computed as matrix-vector products. We show that utilising the Halevi-Shoup method allows the use of rotations and ciphertext packing to scale better compared with the representations in LoLa, when applied to larger convolutional layers. We perform a more detailed investigation on the scalability of the methods used in LoLa to larger models and show that they are significantly outperformed by our proposed method.

- We compare our framework against LoLa by constructing models for MNIST and CIFAR-10. Our main evaluation criteria is the number of HOPs required per inference for a model. With the same layer parameters as LoLa, we are able to obtain over a two-fold reduction in the number of HOPs per inference. Our CIFAR-10 model achieves similar accuracy to that of LoLa's but uses far fewer operations.

## 2 BACKGROUND AND PREREQUISITES

### 2.1 THREAT MODEL

Our threat model concerns that of the machine learning as a service paradigm (MLaaS), in which the user first sends data to a server, which then performs machine learning inference on the received data using some model. The inference result is then delivered back to the user. For example, consider an online machine learning service which claims to detect the probability of a person having COVID-19 from an audio recording of their cough. Suppose that Alice decides to send a recording of her cough to this service, in the hopes of receiving a diagnosis. There are two key threats in this scenario: (i) the risk of an adversary eavesdropping on the data transmission, and (ii) the risk of the MLaaS provider performing unauthorised access on the user's data – in this case, the recording produced by Alice. The first threat can be mitigated using standard cryptographic protocols. However, the second risk is harder to address, especially if the user data is decrypted before inference Bae et al. (2018). The use of HE mitigates both risks. The data is encrypted using HE, which is sufficient to prevent an adversary from eavesdropping. In addition, the provider is only able to perform computations on the encrypted data and will output the inference result without being able to decrypt.

### 2.2 HOMOMORPHIC OPERATIONS

Several recent HE schemes such as BFV (Brakerski & Vaikuntanathan, 2011) and CKKS (Cheon et al., 2017) are based on the RLWE problem and support SIMD ciphertext operations. On a high level, such schemes establish a mapping between real vectors and a plaintext space. The plaintext space is usually the polynomial ring $\mathcal{R} = \mathbb{Z}[X]/(X^N + 1)$. In particular, this is a *cyclotomic polynomial ring* $\mathcal{R} = \mathbb{Z}[X]/(\Phi_M(X))$ where $\Phi_M(X)$ is the $M$-th cyclotomic polynomial and $M = 2N$ is a power of two. The *decoding* operation maps an element in $\mathcal{R}$ to a vector that is either real or complex, depending on the scheme used. The *encoding* operation performs the reverse. Plaintext polynomials are encrypted into ciphertext polynomials using a *public key*. The operations of addition and multiplication can be performed over ciphertexts using an *evaluation key*.

Since each ciphertext corresponds to a vector of real (or complex) values, a single homomorphic operation between two ciphertexts constitutes an element-wise operation between two vectors. In addition, such schemes support rotations of the slots within a ciphertext, with the use of Galois automorphisms.

## 2.3 FLATTENED CONVOLUTIONS

Consider the convolution of an image $\mathbf{I}$ with a filter $f$. For simplicity, assume that both the image and filter are square, the vertical and horizontal strides of the filter are equal. Let $\mathbf{I} \in \mathbb{R}^{d_{\text{in}} \times d_{\text{in}} \times c_{\text{in}}}$, and $f \in \mathbb{R}^{k \times k \times c_{\text{in}}}$. Denote the stride as $s$ and padding as $p$. Now, the output feature map $\mathbf{J}$ is such that $\mathbf{J} \in \mathbb{R}^{d_{\text{out}} \times d_{\text{out}}}$ where

$$d_{\text{out}} = \left\lfloor \frac{d_{\text{in}} - k + 2p}{s} \right\rfloor + 1.$$

A full convolutional layer that outputs $c_{\text{out}}$ feature maps will require a convolution of the input image with each of the $c_{\text{out}}$ filters. Consider the vector $\mathbf{v}$ obtained by flattening each output feature map row-wise. This can be expressed as a matrix-vector product of the form $\mathbf{v} = \mathbf{A} \cdot \mathbf{w} \in \mathbb{R}^{d_{\text{out}}^2 \cdot c_{\text{out}}}$ where $\mathbf{A} \in \mathbb{R}^{d_{\text{out}}^2 \cdot c_{\text{out}} \times d_{\text{in}}^2 \cdot c_{\text{in}}}$ and $\mathbf{w}$ is the flattened representation of $\mathbf{I}$.

## 2.4 FAST CONVOLUTION

The first convolutional layer in a CNN can be represented using convolution-packing (Brutzkus et al., 2019). The convolution of an input image $f$ with a filter $g$ of width $w$, height $h$ and depth $d$ is

$$(f * g)[i, j] = \sum_{x=0}^{w-1} \sum_{y=0}^{h-1} \sum_{z=0}^{d-1} g[x, y, z] f[i + x, j + y, z]. \tag{1}$$

Observe that the parallelism inherent in this computation enables it to be vectorized as

$$f * g = \sum_{x=0}^{w-1} \sum_{y=0}^{h-1} \sum_{z=0}^{d-1} g[x, y, z] \cdot \mathbf{F}^{(x,y,z)}, \tag{2}$$

where $\mathbf{F}^{(x,y,z)}$ is a matrix such that $\mathbf{F}_{ij}^{(x,y,z)} = f[i + x, j + y, z]$. For an input image $I \in \mathbb{R}^{c_{\text{in}} \times d_{\text{in}} \times d_{\text{in}}}$ feature maps and kernel of window size $k \times k$, the input image is represented as $k^2 \cdot c_{\text{in}}$ vectors $\mathbf{v}_1, \ldots, \mathbf{v}_{k^2 \cdot c_{\text{in}}}$, where $\mathbf{v}_i$ contains all elements convolved with the $i$-th value in the filter. Denote corresponding ciphertexts as $\text{ct}_1, \ldots, \text{ct}_{k^2 \cdot c_{\text{in}}}$. The process of producing the $j$-th output feature map is now reduced to ciphertext-plaintext multiplication of each $\text{ct}_i$ with the $i$-th value in the $j$-th filter. In total, the process requires $k^2 \cdot c_{\text{in}}$ ciphertext-scalar multiplications per output feature map, leading to a total of $k^2 \cdot c_{\text{in}} \cdot c_{\text{out}}$ multiplications.

## 2.5 MATRIX-VECTOR MULTIPLICATION

Multiplying a plaintext weight matrix of $\mathbf{A} \in \mathbb{R}^{m \times n}$ with a ciphertext vector can be achieved naively by first performing $m$ ciphertext multiplications of the vector with each row of $\mathbf{A}$. Then for each product ciphertext, we can compute the sum of all elements within it by applying a *rotate-and-sum* procedure (Halevi & Shoup, 2019): we first rotate the ciphertext by $N/2$ slots and add it to the original. Then the procedure is repeated for $N/4$ slots, $N/8$ and so on, until the sum of the ciphertext resides within all slots of the ciphertext. The resulting dot products can be summed together. This basic approach requires $O(m \log n)$ rotations.

Halevi & Shoup (2019) introduced a more efficient method of computing the encrypted matrix-vector product $\mathbf{A} \cdot \mathbf{v}$ for square $\mathbf{A}$. GAZELLE (Juvekar et al., 2018) extended the approach to support rectangular $\mathbf{A} \in \mathbb{R}^{m \times n}$. The method works by decomposing $\mathbf{A}$ into its $m$ diagonals, denoted $\{d_1, d_2, \ldots, d_m\}$ such that $d_i = [\mathbf{A}_{i,0}, \mathbf{A}_{i+1,1}, \ldots, \mathbf{A}_{i+n-1,n-1}]$. Note that all row positions are in modulo $m$. Then each $d_i$ is *rotated* $i$ positions to align values belonging to the same row of $\mathbf{A}$ into the same column(s), and finally each rotated diagonal is multiplied with with corresponding rotations of $\mathbf{v}$. The ciphertexts are summed, and the last stage is to apply a rotate-and-sum procedure to the resulting ciphertext. Overall, this procedure requires $O(m)$ multiplications and $O(m + \log_2 n)$ rotations. We propose an improved variant of this approach for our framework described in section 3.1.

## 3 METHOD

In this section, we present our method for achieving privacy-preserving CNN inference with low numbers of HOPs. In summary, we adopt the fast convolution method from LoLA but compute the intermediate convolutional layers in a network using the Halevi-Shoup (HS) matrix-vector product method instead. This enables large convolutions to be performed in far fewer ciphertext rotations than LoLA's approach of computing a rotate-and-sum procedure for each row in the weight matrix. In section 3.1 we explain the approach we use. In section 3.2, we perform an analysis to show the improvements made by our modifications compared with the approach from LoLA. In section 4, we apply our approach to models for the MNIST and CIFAR-10 datasets.

### 3.1 PERFORMING INTERMEDIATE CONVOLUTIONS

In Section 2.3 we state that a convolution can be flattened and represented as a matrix-vector product $\mathbf{A} \cdot \mathbf{w} \in \mathbb{R}^{d_{\text{out}}^2 \cdot c_{\text{out}}}$ where $\mathbf{A} \in \mathbb{R}^{d_{\text{out}}^2 \cdot c_{\text{out}} \times d_{\text{in}}^2 \cdot c_{\text{in}}}$ and $\mathbf{w}$ is the flattened input.

This product can be computed efficiently using the Halevi-Shoup method. However, neither the original method (Halevi & Shoup, 2019) nor GAZELLE (Juvekar et al., 2018) discuss the constraints the ciphertext slot count $N$ imposes in this context. In Proposition 1 we show that certain constraints must be applied on the sizes of the input and output of the convolution. In particular, we must have $N \geq d_{\text{in}}^2 \cdot c_{\text{in}} + d_{\text{out}}^2 \cdot c_{\text{out}} - 1$, unless $d_{\text{out}}$ and $c_{\text{out}}$ are both powers of 2.

**Proposition 1.** *Let $N$ denote the ciphertext slot count, and $n = d_{in}^2 \cdot c_{in}$ be the size of the convolution input, and $m = d_{out}^2 \cdot c_{out}$ be the size of the output. The basic Halevi-Shoup method, which takes $m - 1 + \lceil \log_2 \left( \frac{m+n-1}{m} \right) \rceil$ rotations, requires the condition that $N \geq m + n - 1$. If this does not hold, but it is the case that $m = 2^l$, $0 \leq l \leq \log_2 N$, then $m - 1 + \log_2 \frac{N}{m}$ rotations are required.*

*Proof.* Applying the Halevi-Shoup technique requires $m$ ciphertext diagonals $\mathbf{d}^{(1)}, \ldots, \mathbf{d}^{(m)}$ of length $n$ to be extracted, rotated and summed. Note that $\mathbf{d}_j^{(i)} = \mathbf{A}_{i+j,j}$. Now, if $N \geq m+n-1$ then $N$ is sufficiently large for all rotations to be performed without wrapping around the ciphertext. If $N < m + n - 1$, then wrap-around will occur for at least one of the diagonal ciphertexts during rotation. For any slot $\mathbf{d}_j^{(i)}$, the rotation by $i$ positions will shift the slot to position $k = i + j \pmod{N}$. The slot $\mathbf{d}_j^{(i)}$ is from the row of $\mathbf{A}$ with index $r = i + j \pmod{m}$, and so must be shifted into an index that is equivalent to $r$ in modulo $m$, in order for the rotate-and-sum algorithm to be used. If $N$ is an integer multiple of $m$, then we see $k \equiv r \pmod{m}$ indeed holds. Otherwise, wrap-around will cause the diagonals to be misaligned when summed together. Since $N$ is a power of 2, the requirement that $m$ divides $N$ is satisfied whenever $d_{\text{out}}$ and $c_{\text{out}}$ are also powers of 2 such that $2 \log_2 d_{\text{out}} + \log_2 c_{\text{out}} \leq \log_2 N$. $\qquad \square$

Based on Proposition 1, we utilise a procedure to ensure that the method can be applied for all choices of $(d_{\text{in}}, d_{\text{out}}, c_{\text{in}}, c_{\text{out}})$ where $d_{\text{out}}^2 \cdot c_{\text{out}} \leq N$: if the condition that $N \geq m + n - 1$ does not hold, then we 'round' $m$ to the closest power of 2 not less than itself, and add corresponding rows filled with 0's to the weight matrix. This procedure is illustrated below for the simple case where $N = 4, m = 3$ and $n = 4$. Notice that the padding allows the diagonals to align correctly after rotation.

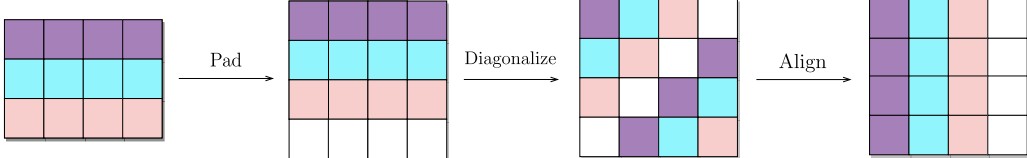

Figure 1: An illustration of our version of the Halevi-Shoup method that utilises a padding method.

### 3.2 COMPARISON WITH LOLA

Firstly, we follow the terminology used by Brutzkus et al. (2019) and define *dense*, *sparse*, *stacked* and *interleaved* representations for a vector $\mathbf{v} \in \mathbb{R}^n$.

- *Sparse*: $\mathbf{v}$ is represented with $n$ ciphertexts, where the $i$-th ciphertext contains the $i$-th element of $\mathbf{v}$ in all its slots.
- *Dense*: $\mathbf{v}$ is represented as a single ciphertext where the first $n$ slots correspond to the values in $\mathbf{v}$.
- *Stacked*: $\mathbf{v}$ is represented as a single ciphertext that contains as many copies of $\mathbf{v}$ as the ciphertext slot count permits.
- *Interleaved*: $\mathbf{v}$ is represented as a single ciphertext similar to the dense representation - however, the slots are shuffled by some permutation.

LOLA proposes a *dense-vector row-major* matrix-vector product method (which we call LOLA-dense) that maps the input vector in the dense representation to an output vector in the sparse representation. Let $\mathbf{A} \in \mathbb{R}^{m \times n}$. The dense-vector row-major method computes $\mathbf{A}\mathbf{v}_1 = \mathbf{v}_2$ via a multiplication per row of $\mathbf{A}$ and summing the elements inside each product vector using the rotate-and-sum procedure. This is the basic approach described in Section 2.5, except the dot products are not summed together.

They also propose a *stacked-vector row-major* method (which we call LOLA-stacked) that requires input vector in the stacked representation and provides the output in the interleaved representation. To obtain the stacked representation, they first pack $k$ copies of $\mathbf{v}$ into a single ciphertext where $k = N/\delta(n)$ and $\delta(n) = 2^{\lceil \log_2 n \rceil}$ is the smallest power of 2 greater than or equal to $n$. The stacked vector is then multiplied with corresponding stacked rows of $\mathbf{A}$, before rotate-and-sum is applied. The main drawback of this method is its reliance on the ability to pack many copies of $\mathbf{v}$ into a ciphertext, which requires $N$ to be much larger than $n$. For large convolutions, this is hard to achieve. We derive the number of operations this method takes.

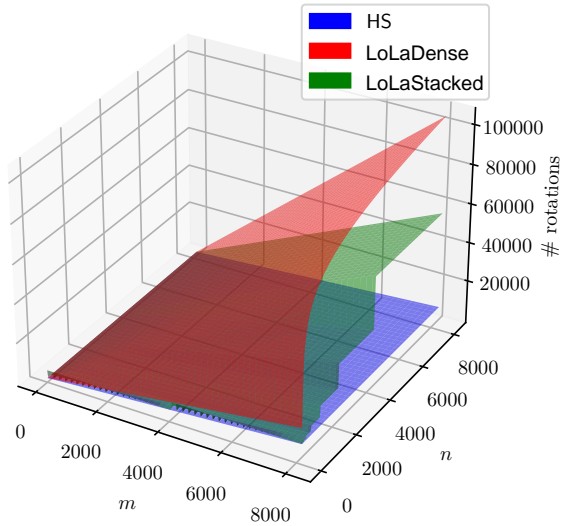

Figure 2: Comparison of LOLA's matrix-vector product methods with the Halevi-Shoup approach, in terms of the number of rotations required for computing a fully-connected layer from $n$ inputs to $m$ outputs.

**Remark 1.** *The* stacked-vector row-major *method proposed by Brutzkus et al. (2019) requires* $\lceil \frac{m}{k} \rceil (k + \lceil \log_2 n \rceil - 1) + \lceil \frac{m}{k} \rceil - 1$ *rotations and* $\lceil \frac{m}{k} \rceil$ *multiplications.*

*Proof.* The stacking in Brutzkus et al. (2019) is done is using $k - 1$ rotations and additions. Then the point-wise multiplication requires a single SIMD multiplication between the ciphertext and $k$

(stacked) rows of $\mathbb{A}$. Finally, $\lceil \log_2 n \rceil$ rotations and additions are performed to compute the $k$ elements of the product vector. Since $N$ is not always sufficiently large to pack $m$ copies of $n$, this procedure has to be performed $\lceil \frac{m}{k} \rceil$ times. To produce all elements of the product vector, $\lceil \frac{m}{k} \rceil (k + \lceil \log_2 n \rceil - 1)$ rotations and $\lceil \frac{m}{k} \rceil$ multiplications are performed, before $\lceil \frac{m}{k} \rceil - 1$ rotations are performed to bring the $\lceil \frac{m}{k} \rceil$ ciphertexts into a single one. Note that the produced ciphertext(s) contain the product elements in an interleaved format, i.e. a permutation. □

Suppose we are passing a 4096-length representation into a fully connected layer to map to a 64-length embedding, and let $N = 16384$. LoLA-dense would require $64 \cdot \log_2 4096 = 768$ rotations and 64 multiplications; LoLA-stacked would require 1023 rotations and 64 multiplications. Now, the Halevi-Shoup product approach would require only $64 + \log_2 8192 = 77$ rotations and 64 multiplications. Since rotations are the most expensive operation, we compare the number of rotations required by each of the three methods in Figure 2.

It can be shown that for any $n > 1$, LoLA-stacked uses fewer rotations than LoLA-dense. However, LoLA-dense can be used to compute two layers instead of one. For large inputs and output layers (relative to $N$), however, both methods require significantly more rotations than HS. LoLA-stacked relies on $m \cdot \delta(n)/N$ being small whereas LoLA-dense relies on $\log_2 n$ being small. With HS, even if $n = N$, $\log_2 n$ is insignificant compared to $m$.

In general, we believe that having the number of rotations be linear to only $m$ is beneficial since neural networks typically down-sample or pool the data to produce denser, higher-level representations, and so can expect $n \geq m$ generally. It should be noted that there are exceptions - such as bottleneck layers.

## 4 EXPERIMENTS

We conduct experiments on the MNIST and CIFAR-10 datasets. We first apply our approach to the CRYPTONETS architecture used in LoLA, to create a model CRYPTONETS-HS. The same architecture was shown to achieve close 98.95% accuracy by Gilad-Bachrach et al. (2016), and we observe similar performance using their training parameters. Training is conducted using TensorFlow (Abadi et al., 2015). The architecture is then converted into a sequence of homomorphic operations. We use the SEAL library (Chen et al., 2017) for this. For reference, the original CRYPTONETS architecture is shown in Figure 3.

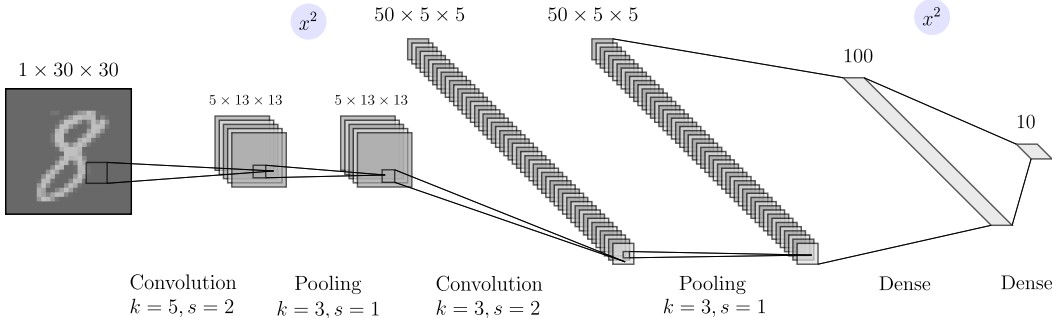

Figure 3: The architecture used in LoLa. $k$ indicates the kernel size and $s$ indicates the stride of convolution and pooling. Note that the original $28 \times 28$ input is padded.

LoLA uses a combination of their stacked and interleaved representations for their intermediate convolutions. We opt to use the efficient approach instead. To reduce consumption of instruction depth, linear layers without activations between them are composed together. For instance, each convolution-pooling block of CRYPTONETS-HS is a single linear layer. This is done to enable our baseline to match the architecture in LoLA. It should be noted that omitting activations can reduce representational power and made not be ideal in practice. The models are implemented using operations provided by the SEAL library, and inference is performed on a standard desktop

processor. We utilise only a single thread. We measure the number of homomorphic operations required per single inference, as well as model test accuracy.

Using the proposed improvements, we also construct models ME and CE, for MNIST and CIFAR-10, respectively. The proposed architectures have reduced memory requirements and layer parameters that are better suited towards the HS method:

- Model ME has similar test accuracy to CRYPTONETS-HS but is designed in consideration of the way we are computing the layers. We note that applying the HS method in computing layer 6 of CRYPTONETS-HS requires setting $m = 100$ and $n = 845$. Specifically, we reduce the kernel size of the first convolution from $5 \times 5$ to $3 \times 3$, and the size of the first dense layer from 100 to 32. To account for the reduced representational power of the first convolution, the stride is reduced from 2 to 1. This notably does not add any homomorphic operations to the computation. We are able to achieve 98.7% test accuracy using ME after 100 epochs of training with the Adam optimiser.

- Model CE is larger, and requires more operations to compute. We initially construct a model with a second convolutional layer with an output tensor of size $8 \times 8$ with 64 channels, and then follow the approach from Lu et al. (2021) and use SVD to factorise this layer into a smaller sub-convolution followed by a $1 \times 1$ convolution which is efficient to compute using the convolution-packing method described in LoLa. The smaller sub-convolution is computed using the HS method. The final trained model has a test accuracy of 73.1%.

| Layer | Description | Parameters | Input |
|---|---|---|---|
| 1 | Convolution | $k = 3, s = 1$ | (1, 30, 30) |
| - | Square | - | (5, 28, 28) |
| 2 | Avg. Pool | $k = 3, s = 2$ | (5, 28, 28) |
| 3 | Convolution | $k = 3, s = 1$ | (50, 14, 14) |
| 4 | Avg. Pool | $k = 3, s = 2$ | (50, 12, 12) |
| 5 | Flatten | - | (50, 12, 12) |
| 6 | Dense | $m = 32$ | (1250) |
| - | Square | - | (32) |
| 7 | Dense | $m = 10$ | (32) |
| - | Softmax | - | (100) |

(a) ME layer parameters.

| Layer | Description | Parameters | Input |
|---|---|---|---|
| 1 | Convolution | $k = 3, s = 1$ | (1, 32, 32) |
| - | Square | | (18, 30, 30) |
| 2 | Average Pooling | $k = 2, s = 2$ | (18, 30, 30) |
| 3 | Sub-convolution | $k = 3, s = 1$ | (18, 10, 10) |
| 4 | Sub-convolution | $k = 1, s = 1$ | (13, 8, 8) |
| - | Square | | (18, 30, 30) |
| 5 | Average Pooling | $k = 2, s = 2$ | (64, 8, 8) |
| 6 | Convolution | $k = 3, s = 1$ | (64, 4, 4) |
| 7 | Flatten | - | (256, 2, 2) |
| 8 | Dense | $m = 512$ | (1024) |
| - | Square | | (18, 30, 30) |
| 9 | Dense | $m = 10$ | (256) |
| - | Softmax | - | (10) |

(b) CE layer parameters.

Table 1: Layer parameters for ME (left) and CE (bottom). $k$ and $s$ indicate the kernel width and stride respectively. $m$ indicates the number of output nodes of a fully-connected layer.

# 5 RESULTS AND DISCUSSION

Table 2 shows a breakdown of the operations for the models discussed so far. In homomorphic encryption applied to neural networks, the most expensive homomorphic operation is rotation, with a worst case time complexity of performing both a number theoretic transform (NTT) and an inverse

NTT on vectors of length $N$. We observe that CRYPTONETS-HS requires a total of 122 rotations, as shown in Table 2a. LOLA-MNIST requires a total of 380 rotations[1] for the same architecture. We notice that at both the convolution-pooling blocks in the CRYPTONETS architecture, the HS implementation has fewer rotations than LOLA. The reduction in rotations is a direct result of applying the HS method to compute flattened convolutions, in which the number of rotations scales with the size of the output tensor, rather than the size of the input. LOLA uses the stacked-vector row-major method for the first convolution, and the dense-vector row-major method for the second convolution. In Section 3.2 we discussed that both of these scale poorly compared to the HS method.

Now, the optimised ME architecture requires only 56 rotations per inference, but still achieves a high test accuracy of 98.7%, which is close to the 98.95% test accuracy achieved by LOLA-MNIST, suggesting that careful selection of layer sizes can lead to great reductions in inference latency. We do note that CRYPTONETS-HS requires more multiplications than LOLA-MNIST (245 vs. 148). This is due to the stacked-vector row-major approach reducing the number of multiplications needed. However, the reduction in rotations exceeds the increase in multiplications.

| Layer | Total HOPs | | | Add PC | | | Add CC | | | Mul PC | | | Mul CC | | | Rot | | |
|---|---|---|---|---|---|---|---|---|---|---|---|---|---|---|---|---|---|---|
| | $M$ | $L'$ | $L$ | $M$ | $L'$ | $L$ | $M$ | $L'$ | $L$ | $M$ | $L'$ | $L$ | $M$ | $L'$ | $L$ | $M$ | $L'$ | $L$ |
| Conv1 | 90 | 250 | 250 | 5 | 5 | 5 | 40 | 120 | 120 | 45 | 125 | 125 | - | - | - | - | - | - |
| Flat1 | 8 | 8 | 8 | - | - | - | 4 | 4 | 4 | - | - | - | - | - | - | 4 | 4 | 4 |
| Square1 | 1 | 1 | 1 | - | - | - | - | - | - | - | - | - | 1 | 1 | 1 | - | - | - |
| Conv2-Dense1 | 110 | 308 | 492 | 1 | 1 | - | 38 | 103 | 246 | 32 | 100 | 13 | - | - | - | 39 | 104 | 246 |
| Square2 | 1 | 1 | 1 | - | - | - | - | - | - | - | - | - | 1 | 1 | 1 | - | - | - |
| Dense2 | 36 | 38 | 279 | 1 | 1 | - | 12 | 13 | 139 | 10 | 10 | 10 | - | - | - | 13 | 14 | 130 |
| Total | **246** | 606 | 1031 | 7 | 7 | **5** | **94** | 240 | 509 | **87** | 235 | 148 | 2 | 2 | 2 | **56** | 122 | 380 |

(a) Comparison of operations in three different privacy-preserving MNIST models. $L$ indicates the original LOLA-MNIST model, $L'$ indicates the CRYPTONETS-HS model, and $M$ indicates the ME model. 'CC' indicates an operation between two ciphertexts and 'PC' indicates an operation between a plaintext and a ciphertext.

| Layer | HOPs | Add PC | Add CC | Mul PC | Mul CC | Rot |
|---|---|---|---|---|---|---|
| Conv1 | 972 | 18 | 468 | 486 | - | - |
| Flat1 | 30 | - | 15 | - | - | 15 |
| Square1 | 3 | - | - | - | 3 | - |
| Pool1-Conv2 | 7506 | 1 | 2504 | 2496 | - | 2505 |
| Conv3 | 1677 | 64 | 768 | 832 | - | 13 |
| Flat2 | 126 | - | 63 | - | - | 63 |
| Square2 | 1 | - | - | - | 1 | - |
| Pool2-Dense1 | 778 | 1 | 260 | 256 | - | 261 |
| Square | 1 | - | - | - | 1 | - |
| Dense2 | 40 | 1 | 14 | 10 | - | 15 |
| Total | 11134 | 85 | 4092 | 4080 | 5 | 2872 |

(b) Break-down of operations in CE.

Table 2: Break-down of the types of operations performed by the low latency models. The notation 'layer1-layer2' denotes the layers between and including layer1 and layer2.

In terms of latency, the CRYPTONETS-HS model requires 2.7 seconds, whereas ME requires only 0.97 seconds. For reference, LOLA-MNIST requires 2.2 seconds per inference; however, they utilise 8 cores on a server CPU whereas our homomorphic operations are run on a single core of a desktop processor.

For the CE model, we provide a model with sightly lower test accuracy (73.1% vs. 74.1%) but requiring less than 10% the number of rotations than LoLa's CIFAR-10 model. This due to both ensuring that intermediate tensor sizes fit well into the number of available slots, and also the use of the HS method. The number of operations for each layer is shown in 2b. It should be noted that a sizeable portion of this reduction comes from using a smaller model. The exact model from LOLA

---

[1]Note that this is deduced to the best of our ability using the descriptions supplied in their paper.

was not implemented due to its high memory requirements. However, we expect that utilising our approach for LOLA's CIFAR-10 architecture would still significantly reduce the inference latency.

The main takeaway from the results is that the application of the HS method to the framework proposed by LOLA can significantly reduce the number of rotations, which are the most computationally expensive HE operation. In addition, our models ME and CE are more efficient than the ones proposed by LOLA whilst achieving similar accuracy.

## 6 CONCLUSION

Privacy-preserving inference using homomorphic encryption is largely constrained by the computational requirements of the operations. We propose improvements over LOLA to achieve lower latencies computing intermediate convolutions, resulting in over a two-fold reduction in the number of rotations for the same MNIST architecture. It is clear that further improvements can made along this direction, especially in the topic of automatically selecting suitable layer parameters to set the trade-off between inference latency and model accuracy (Lou et al., 2020).

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
