# OpenReview forum: "Efficient representations for privacy-preserving inference"
_ICLR.cc/2022/Conference — ICLR 2022 Submitted_

### Official Review · Reviewer_gqrN · 2021-11-01

**Correctness:** 4
**Technical Novelty And Significance:** 3
**Empirical Novelty And Significance:** 3
**Recommendation:** 8
**Confidence:** 3

**Main Review:**

The technique is explained well and there is an-depth analysis of the resulting complexity with a comparison to prior work. The 2-3x improvement is considerable. Furthermore, the editorial quality is high.

Minor:
- p7: 56 rotation(s)


**Summary Of The Paper:**

The paper applies the state-of-the-art technique for matrix-vector multiplication in homomorphic encryption to encrypted inference for convolutional neural networks.


**Summary Of The Review:**

The paper provides a well-argued application of the state of the art in homomorphic encryption.

---

> ### Author Response · Authors · 2021-11-20
> **Thank you for the review**
>
> We would like to thank the reviewer for their insightful comments and feedback. The minor error that was pointed out has been corrected.

---

### Official Review · Reviewer_uGUc · 2021-11-04

**Correctness:** 4
**Technical Novelty And Significance:** 2
**Empirical Novelty And Significance:** 2
**Recommendation:** 6
**Confidence:** 3

**Main Review:**

Overall, this paper addresses an important problem in privacy-preserving machine learning and the results show that the method is effective. However, the originality and the evaluation of the paper are not strong enough. The originality is incremental because the authors applied an existed approach, HS, to an existed structure, CRYPTONET. The evaluation is not sufficient because the experimental results are not well analyzed and interpreted. For example, only a limited part in table 2 is discussed in the text. More analysis about table 2 would improve the results section of the paper. In addition, the introduction to the HS method is a little short and the entire pipeline of the HS approach is not easy to follow. By focusing on the introduction to the HS method instead of the introduction of the baseline LOLA method, the authors can explain the HS method better.

**Summary Of The Paper:**

To accelerate the privacy-preserving inference through convolution neural networks (CNNs) with homomorphic encryption (HE), the authors aim to reduce the number of homomorphic operations (HOPs) required for the algorithm to save the data needs to be transferred while preserving prediction accuracy.

Using a LOLA method as a baseline, the authors used Halevi-Shoup (HS) which requires much fewer rotations operations in general. In the experimental results, the HS method requires half of the operations in comparison to the LOLA method. By further simplifying the structure of the network, only about 20% of HOPs are required. These simplifications bring about 1% and 27% accuracy loss.


**Summary Of The Review:**

The paper is marginal acceptable because the paper is addressing an important question and the method is effective. However, the improvement of the method is not impressing or suppressing. The entire is not well finished. Thus, this paper is regarded as a border line paper.

---

> ### Author Response · Authors · 2021-11-20
> **Thank you for the review**
>
> > The originality is incremental because the authors applied an existed approach, HS, to an existed structure, CRYPTONET.
>
> We agree that our method combines existing ideas. However, we believe that the gain in performance is more than incremental compared to LoLa. The methods used by LoLa scale poorly for larger models and this has been addressed by our work to a great extent as we achieve more than 2 x reduction in operation count. From our analysis of the asymptotic behaviour of the methods we show that our approach considerably outscales theirs.
>
> > The evaluation is not sufficient because the experimental results are not well analyzed and interpreted. For example, only a limited part in table 2 is discussed in the text. More analysis about table 2 would improve the results section of the paper.
>
> We believe the main take-away point from the results is the reduction in operations resulting from the use of our proposed approach over the baseline, and the fact that our models with similar accuracy requires much less computation. We have tried to make these points clearer in our interpretation of the results.
>
> >  the introduction to the HS method is a little short and the entire pipeline of the HS approach is not easy to follow. By focusing on the introduction to the HS method instead of the introduction of the baseline LOLA method, the authors can explain the HS method better.
>
> Thank you for highlighting this - we expanded the introduction to the HS method in Section 2 and also made it more explicit what changes we made to the method in Section 3, which itself has been rewritten partially to be more coherent.

---

> > ### Comment · Reviewer_uGUc · 2021-11-29
> > **Response**
> >
> > I have read the authors response and stand by my recommendation.

---

### Official Review · Reviewer_9nHn · 2021-11-06

**Correctness:** 2
**Technical Novelty And Significance:** 1
**Empirical Novelty And Significance:** 1
**Recommendation:** 3
**Confidence:** 5

**Main Review:**

Pros:

1. The paper attempts to study a problem that is at the heart of privacy preserving machine learning, i.e., speeding up HE based inference on deep neural networks.
2. The paper is well written.

Cons:

1. The paper is rather straightforward rehashing of existing (which are themselves rather simplistic, in my opinion). For instance, they take a recent improvement by Halevi and Shoup for Matrix products and apply it to the previous work LoLa.

2. The experimental section is quite weak and misleading. In the abstract it is claimed that the current work improves upon the existing work by two orders of magnitude. While this is true when compared by CryptoNets -- the more recent works already seem significantly better.



**Summary Of The Paper:**

The paper considers the problem of privacy preserving inference on deep learning models using homomorphic encryption. HE is a special type of encryption that allows one to perform certain types of computations while the data is encrypted. However, the catch is HE based inferences can be significantly slower than the non-private counterparts. The paper claims to improve upon the existing state of the art HE based inference approaches significantly -- two orders of magnitude.

**Summary Of The Review:**

I think this is an important problem and the paper is reasonably well written. Unfortunately, it is a rather weak attempt. Essentially, it is a rehashing of existing ideas -- which are already uninspiring from an ML point of view. Further, as described above, the experimental section is rather misleading. For instance, there is no clear comparison of inference time with LoLa (does it improve the inference time at all?).

---

> ### Author Response · Authors · 2021-11-20
> **Thank you for the review**
>
> We would like to thank the reviewer for their time and their insightful comments. Below are our responses.
>
> > The paper is rather straightforward rehashing of existing (which are themselves rather simplistic, in my opinion).
>
> We believe that though our approach is simple, it is still effective and improves previous works by a large margin of more than 2 x reduction in operation count. We would like to express our belief that simplicity alone is not a good reason to reject a paper, and that a simple, effective solution can be a more valuable contribution than an effective yet convoluted one.
>
> > The experimental section is quite weak and misleading. In the abstract it is claimed that the current work improves upon the existing work by two orders of magnitude. While this is true when compared by CryptoNets -- the more recent works already seem significantly better.
>
> We would like to kindly point out to the reviewer that we never claimed to give two orders of magnitude of improvement. The abstract mentions ‘two-fold reduction’ (which is equivalent to saying 50% reduction), not two orders of magnitude. The claim of ‘two-fold reduction’ is supported by our results. We hope that the reviewer can reconsider their stance that the claims we make are misleading.
>
> > there is no clear comparison of inference time with LoLa (does it improve the inference time at all?).
>
> We would like to highlight the fact that one of the main contributions of our work is that we analytically prove the number of operations of our models and also that of previous work. In this way, we are able to perform a fair comparison without being affected by discrepancies in hardware. We believe the alternative approach of measuring the time latencies would provide little insight due to the direct relationship between the number of operations required and the time taken. Previous work often compared running times given by different authors, even though the times were measured with different hardware systems. We believe our approach is much fairer and also easier to apply.

---

> ### Author Response · Authors · 2021-11-26
> **Discussion Period Ending - Final Questions / Comments**
>
> Dear Reviewer,
>
> We would again like to thankyou for your time in reviewing our manuscript and the useful feedback. We would like to highlight that the discussion period is coming to an end and were hoping you might be able to respond to our comments and consider re-evaluating the score/confidence.
>
> In particular we would like to re-highlight that you have misread our claims regarding improvements. We never claim a 2 orders of magnitude improvements as you have stated in your review, we only claim 2x improvement which differs to 10^2 (2 orders of magnitude). We believe it is very important that this is addressed as one of the main reasons you cite as motives to reject are:
>
> >  In the abstract it is claimed that the current work improves upon the existing work by two orders of magnitude  ..
> > The experimental section is .../misleading
> >  Several of the paper’s claims are incorrect
>
> Thus we would kindly ask if its possible to re-evaluate/retract these particular notes which state our claims are misleading/incorrect. Another important point on this matter is we never claim to be SOTA among the latest work, our claims are quite clear:
>
> > We build on on top of certain methods that we compare to and we prove that we improve upon said methods that we augmented.
>
> Thus we are not making any misleading claims, the claims are self contained and precise.
>
> Regarding experimental weakness: in this work we carefully derive the number of operations of all methods we consider. Prior work does not provide openly these calculations/derivaions and in some cases do not provide these results/final formulas. We use these formulas to carefully compare to previous work, this is a rigorous and carefull comparison (independant of hardware) and it is clear that this would be proportional to comparing the running times on the same architectures. This is not the first work in this area to carry out their main comparisons based on the number of operations, thus we would like to ask that you carefully reconsider your stance on the weakness of the experiments and lack of comparison to LoLa.
>
> Finally, there are also points regarding the simplicity/combinations of methods as a reason to reject in the main response. We believe that this is a core point that merits discussion and needs addressing, specially since most PMLR (including ICLR) and ACL venues highlight that "the papers method being too simple" is an invalid reason to reject, sources:
>
> 1. https://sites.umiacs.umd.edu/elm/2016/02/01/mistakes-reviewers-make/ (this is linked directly from the ICLR reviewing guidelines https://iclr.cc/Conferences/2021/ReviewerGuide)
> 2. https://2020.emnlp.org/blog/2020-05-17-write-good-reviews (Linked directly from the ACL review guidelines)
>
> It is also unclear and difficult to understand what "uninspiring from an ML point of view" means when criticising simplicity, the word unispiring seems unprecise and subjective and ML being a very broad multidisciplinary field makes the claim ambigous (there are many different ML points of views that arise from the varied backgrounds in the field), thus a clarification/detailed point would be extremely helpful here.
>
> We would really appreciate your response as it is clear some very core/key details have been misread and misquoted in this review and we believe it is very important to correct these and consider re-evaluating scores and confidences appropiately.
>
> P.S. We thank you for your time in reviewing our paper and reading our responses, and look forward to hearing from you.
>
> Regards,
> Paper3604 authors

---

### Official Review · Reviewer_7wWC · 2021-11-08

**Correctness:** 3
**Technical Novelty And Significance:** 1
**Empirical Novelty And Significance:** Not applicable
**Recommendation:** 5
**Confidence:** 2

**Main Review:**

**Strengths**

The experimental results demonstrate that the proposed method is much more efficient than previous works.


**Weaknesses**

Disclaimer: The reviewer is not familiar with the literature of HE schemes for privacy-preserving inference, and has not checked the relevant literature closely.

Overall, the reviewer feels that the writing of this paper is not clear, and many details are missing or not easy to follow. For example, the main methods are supposed to be in Section 3.1, while this section only contains a primitive introduction to the convolution operation and its flattened version (which I would expect to reside in Section 2), and a remark which tells the number of operations for the proposed method. The two parts are a bit disconnected. Based on my understanding, the proposed method is to apply the HE operation (as in Section 2.4) on the flattened convolution operation, and this part should go before the remark part. But the corresponding description is completely missing from my point of view.

In section 3.2, the authors compared with two types of representations in LoLa: the sparse representation and the stacked representation. I roughly checked the LoLa paper (Brutzkus et al., 2019). Actually they have five types of representations, even including a "Convolution representation" which claims to "make convolution operations efficient". I didn't check the details of their "Convolution representation", but I would like to ask the authors what is the difference between their method and yours, and why you didn't involve a comparison in your paper? I would appreciate it if the authors can provide a clear explanation to this.

In experiments, the authors mentioned that "linear layers without activations between them are composed together". But activations are often quite important in all sorts of architectures. Is there any reason that impeded authors from using activations? Also, the last column of Table 1(a) (i.e., the "Input" column) look wrong to me. I would suggest that the authors carefully check the numbers.

Minors:

* Authors said "We choose to use a variant of this method ... ". So is the variant proposed by the authors? What is the relationship of this variant and Halevi & Shoup (2019) and GAZELLE (Juvekar et al., 2018)?
* In Section 3.1, $\mathbf{v}$ should be $\in \mathbb{R}^{d_{\mathrm out}^2\cdot c_{\mathrm{out}}}$
* In Section 3.1, the "Remark" should be replaced by "Proposition".

**Summary Of The Paper:**

This paper follows the line of work that leverages holomorphic encryption (HE) operations on encrypted data for privacy-preserving inference on MLPs and CNNs. Authors propose changes to the representations of intermediate layers during CNN inference, which requires much fewer number of operations than previous works.

**Summary Of The Review:**

Several important details and comparisons are missing in the paper, which largely weakens the contribution of the paper. Also, there are flaws in the statement, and the writing is not professional enough.

---

> ### Author Response · Authors · 2021-11-20
> **Thank you for the review**
>
> We would like to thank the reviewer for their insightful feedback and suggestions for improvement.
>
> We re-organised the structure of our paper to address the concerns made. In particular, we have added more detail to section 3.1 in order to clarify our main method. The description of convolution has been moved to section 2. Several parts of Section 3 have been re-written as it was previously unclear with respect to the descriptions of the ‘representations’, which we used to refer to the way in which LoLa computes the matrix-vector products. The sparse representation refers to the dense-vector row-major method from LoLa, and the stacked representation refers to the stacked-vector row-major method. We have now reworded it to use the same terminology as the LoLa paper.
>
> > I didn't check the details of their "Convolution representation", but I would like to ask the authors what is the difference between their method and yours
>
> The convolution representation is mentioned Section 2 of our paper - it is only used for computing the first convolution of the model. We describe its role in our approach at the start of Section 3.
>
> > In experiments, the authors mentioned that "linear layers without activations between them are composed together". But activations are often quite important in all sorts of architectures. Is there any reason that impeded authors from using activations?
>
> Thank you for pointing this out. We use consecutive layers without intermediate activations to match the original CryptoNets architecture and the one used by LoLa, in order for our comparison to be fair. We have added a comment in the paper to address the fact that not having non-linear activations between layers does not add representational power to the model and may not be a sensible choice of architecture in practice. In the context of HE, using non-linear activations is expensive because it consumes one level of depth in the computation.
>
> >Also, the last column of Table 1(a) (i.e., the "Input" column) look wrong to me. I would suggest that the authors carefully check the numbers.
>
> We have double checked the table values and found no errors. We want to clarify that this table is not the same model as the CryptoNets model.
>
> >"We choose to use a variant of this method ... ". So is the variant proposed by the authors? What is the relationship of this variant and Halevi & Shoup (2019) and GAZELLE (Juvekar et al., 2018)?
>
> We have made the description of the use of Halevi & Shoup’s method clearer. In summary, our variant is similar to the original and the one by Juvekar et al but we introduce an improvement. We allow the method to work for all possible flattened convolution as long as the input and output sizes do not exceed the ciphertext capacity. This required an analysis of the constraints on input and output sizes, which we provide in Section 3.1.

---

> > ### Comment · Reviewer_7wWC · 2021-11-24
> > **Thank you for the response**
> >
> > I have read the response and the updated revision. My concerns are adequately addressed and the paper structure/presentation has substantially improved. Thank you for the effort. I have increased my score to 5, but again I am not an expert in this field, and other reviewers' comments shall count more towards the final decision.
> >
> > Best regards,
> >
> > Reviewer 7wWC

---

> > > ### Author Response · Authors · 2021-11-24
> > > **Thank you for the update and the constructive feedback.**
> > >
> > > Dear Reviewer,
> > >
> > > Thank you for the constructive feedback and the very timely update. We would like to highlight that your comments significantly increased the readability and quality of the manuscript. We notice that your new score is still tending towards a rejection (marginally below the acceptance threshold) and are wondering if there are any potential clarifications, improvements, or tips that you think would push this manuscript over towards the acceptance region, is there something we were not able to address that we could clarify further?
> > >
> > > As you mentioned in the original review, we carefully show in our experiments that our proposed approach significantly reduces the number of calculations (which will directly reduce the running time of the methods) compared to previous works which we believe to be a useful contribution thus it would be very helpful to know how to progress/improve this manuscript further.
> > >
> > > Another important contribution that we would like to highlight is that we compare the schemes in question directly on the number of operations they take (which required proof), circumventing the need for measuring running time empirically as this depends on the hardware architecture which the methods are run on and thus does not allow for a direct comparison between different results across papers without re-running all methods under the same environment.
> > >
> > > Kind regards,
> > > Authors of Paper 3604

---

### Author Response · Authors · 2021-11-22
**Summary of updates**

Following the reviews, we have updated the manuscript with the following changes:
* Attempted to make the writing clearer, addressing the issues pointed out by some of the reviewers. We have tried to make the explanation of previous work LoLa clearer in Section 3.2, and clarified our the description of the method in Section 3.1, which has a new figure added.
* Added more discussion of the experimental results in Section 5.
* Fixed minor typos spotted by the reviewers.

Overall despite the approach of our work being relatively simple, it still brings strong improvements over previous work. This can be seen through the models we implemented and the comparisons made. We believe it is not a bad thing that the improvements are simple yet effective. We also make the contribution of analyzing the performance of previously proposed methods from LoLa in greater detail than the original paper. We believe that basing our comparisons around operation usage is a better method to measuring time differences in benchmarks, due to the large discrepancies in the hardware used to produce results.

---

### Author Response · Authors · 2021-11-26
**Final Questions**

Dear reviewers,

As the end of the discussion is approaching, we would like to kindly ask you to consider our responses to your concerns. We answered each of your questions, and updated the manuscript based on your feedback. Thank you for engaging into the discussion so far, and please let us know if you have any final questions that we can address.

Authors

---

### Decision · Program_Chairs · 2022-01-20

**Decision:**

Reject

**Comment:**

This paper improves on the efficiency of prior work that uses homomorphic
encryption to perform privacy-preserving inference. There are two main
concerns raised by the reviewers. First, multiple reviewers (and I) found
this paper difficult to read. Multiple pieces of the problem are not
clearly presented especially with respect to the technical contributions.
This was fixed in part in the rebuttal but more could still be done here.
But more importantly, three reviewers raise concerns about the evaluation
methodology, especially with respect to comparisons to prior work. On top
of this, there are valid criticisms raised by the reviewers about if the
contribution here is that significant when compared to prior work. (This
is something that both more clear writing and more careful experiments
could hep address.) Taken together I do not believe this paper is yet ready
for publication.